# Management of Invasive Infections due to a Rare Arthroconidial Yeast, *Saprochaete capitata*, in Two Patients with Acute Hematological Malignancies

**DOI:** 10.3390/vaccines9111289

**Published:** 2021-11-06

**Authors:** Francesca Gurrieri, Silvia Corbellini, Giorgio Piccinelli, Alessandro Turra, Enrico Morello, Michele Malagola, Domenico Russo, Arnaldo Caruso, Maria Antonia De Francesco

**Affiliations:** 1Department of Molecular and Translational Medicine, Institute of Microbiology, University of Brescia-ASST Spedali Civili, 25123 Brescia, Italy; francescagurrieri@me.com (F.G.); silvia.corbellini@gmail.com (S.C.); giorgio.piccinelli@gmail.com (G.P.); arnaldo.caruso@unibs.it (A.C.); 2Chair of Hematology, Unit of Blood Diseases and Stem Cells Transplantation, University of Brescia-ASST Spedali Civili, 25123 Brescia, Italy; alessandroturra@ymail.com (A.T.); enrico.morello@asst-spedalicivili.it (E.M.); michele.malagola@unibs.it (M.M.); domenico.russo@unibs.it (D.R.)

**Keywords:** immunosuppression, fungal infection, neutropenia, bone marrow transplantation

## Abstract

*Saprochaete capitata* is an arthroconidial yeast, found principally in the environment, even if it belongs also to the normal microbial flora that colonize human subjects. This yeast is increasingly associated with invasive infections in hematological patients, in particular in those affected by acute leukemia. An important risk factor that predisposes to this infection is the profound neutropenia present in such immunocompromised patients. *Saprochaete* spp. were found resistant to both echinocandins and fluconazole so the treatment is often difficult. Here, we report two cases of sepsis in two patients with acute leukemia. All of them had fatal events, due to the worsening of their clinical condition. An early diagnosis and appropriate management of these pathogens is important in consideration of the poor prognosis associated to these fungal invasive infections.

## 1. Introduction

Invasive systemic fungal infections have been recognized to cause life-threatening infections, particularly in the setting of immunosuppression [1,2,3].

Until now, *Candida* species were the most frequent fungi isolated in immunocompromised subjects and the third most prevalent cause of bloodstream infection acquired in hospital environments [4]. In the last few years, however, other fungi have emerged as pathogens. These rare yeasts are associated with high mortality rate [5,6,7,8]. The predisposing conditions are related to impaired immune defenses due to hematological neoplasia, transplantation, neutropenia and HIV infection. These invasive infections are furthermore associated to the treatment with anticancer or immunosuppressant drugs, exposure to antifungal agents, the use of catheters, the presence of comorbidities and interference of antibiotics with the commensal flora [9,10,11,12,13,14].

Correct antifungal prophylaxis is very important to reduce mortality; the intrinsic resistance that these rare yeasts exhibit against one or more classes of antifungals [15,16,17,18], however, can complicate the achievement of this goal. The antifungal susceptibility pattern of these fungi is difficult to interpret for the absence of clinical breakpoints. For this reason, targeted treatment that takes into account the species level of the fungus and susceptibility profiles is imperative for clinical management.

*Saprochaete capitata*, formerly named *Geotrichum capitatum, Blastoschizomyces capitatum*, *Magnusiomyces capitatum*, is a rare emerging fungus responsible for invasive diseases in hematological patients affected by severe neutropenia or in patients who underwent hematopoietic stem cell transplantation [19,20].

It is classified in the family Dipodascaceae that belongs to the order Saccharomycetales and it is associated with ascomycetous yeast, even if its tallus is filamentous without budding cells [21]. It produces typical hyphae and arthroconidia, which are rectangular or rounded at the ends. It is urease negative and grows well at 25 °C and in the presence of cycloheximide.

The first report of systemic infection with *Saprochaete capitata* was reported in 1964 [22]; after this date different opportunistic infections due to this fungus were described.

Sporadic cases and outbreaks of fungemia due to *Saprochaete capitata* have been reported principally in Europe and in particular in Italy [23,24,25]. 

Mazzocato S et al. identified a total of 104 cases of systemic infections due to *S. capitata* described between 1977 and 2013 [26]. Fifty-six percent of the patients were male and the median age was 56 years (range 0.5–76 years). Hematological malignancies were found to be the most common comorbidities: Acute Myeloid Lekaemia (AML) (52%), Acute Lymphoblastic Leukaemia (ALL) (22%) and other hematological malignancies (13%). Other underlying diseases (COPD, diabetes, solid tumors, solid organ transplantation and endocarditis) were present in 9% of the cases, or there were no reported diseases. Severe neutropenia (<500/mm^3^) was present in 82% of the patients.

A case of *Saprochaete capitata* infection has also been reported in a patient with CARD9 deficiency—a genetic immune disorder characterized by susceptibility to fungal infections [27]. 

This fungus is diffuse in the environment, especially in soil [28,29]; it was found in dairy products and in household dishwashers. The entry of the yeast in the host is not well known; however, respiratory or gastrointestinal involvement is probable [23,24,25]. Eventual disruption of gastrointestinal mucosa and ulceration determined by chemotherapy may induce a passage of the yeast in the blood, increasing the risk of fungemia [23].

The diagnosis of *Saprochaete* invasive infections is made observing the presence of typical arthroconidia by microscopy from sterile sites. However, *Saprochaete capitata* cannot be differentiated from *Saprochaete clavata* either by macroscopic or microscopic analysis [30]. Molecular analysis, such as nucleotide sequencing of internal transcribed spacer (ITS) and partial large subunit (LSU) or multilocus sequencing, might allow more precise discrimination [25,30,31]. Furthermore, also MALDI-TOF mass spectrometry is a useful and reliable method able to identify the different athroconidial fungi [32]. 

The distinction of the two fungal species is important not only for epidemiological studies, but also because they differ in antimicrobials susceptibility patterns. 

Despite the start of an adequate antifungal therapy, mortality rate due to invasive *Saprochaete* spp. infections ranges from 40 to 89% in some patient groups.

Here, we report two cases of bloodstream infections due to *Saprochaeate capitata* both in patients treated with intensive chemotherapy for acute leukemia and as a complication post allogeneic bone marrow transplantation.

Case 1. In July 2018, a 52-year-old woman was referred to the Hematology Unit of the Hospital Spedali Civili of Brescia for a severe leukocytosis sustained by monoblastic cells. She was affected by multiple sclerosis. Bone marrow aspiration was performed and a diagnosis of acute myeloid leukemia (AML) was made. Immunophenotyping by flow cytometry detected 80% of blast cells expressing myeloid markers CD34+, CD117−, CD33+/−, HLADR−, CD11b+, CD5+ and CD7+ with point mutations in NRAS, U2AF1, DNMT3A and IDH2 genes. Therefore, induction chemotherapy (Idarubicine/Cytarabine/Etoposide) was started, followed by four consolidation cycles based on high-dose cytarabine. The disease entered complete remission after induction and it was maintained until October 2019. In October 2019, the disease relapsed and reinduction treatment with fludarabine/cytarabine/idarubicin was started without complete remission achievement and with neutropenia and thrombocytopenia (0.36 × 10^3^ and 32 × 10^3^/μL, respectively). In January 2020, enasidenib was started (100 mg/daily). After one month of absence of remission, the therapy was changed with the addition of azacitidine (75 mg/sqm/daily for 7 days on a 28 day-cycle). In March 2020, she was admitted at our unit for the presence of fever, nausea and vomitus. Blood cultures were positive for *Staphylococcus haemolyticus* and *Pseudomonas aeruginosa.* Antibiotic therapy with ceftolozane/tazobactam (1 g/0.5 g/every 8 h for 3 weeks), colistin (9 × 10^6^ U/three times a day for 2 weeks) and clyndamicin (300 mg every 6 h for 4 weeks) was started and the central venous catheter replaced. The patient left the unit on 30 April 2020.

In June 2020, she was readmitted at the Hematology unit for bone marrow transplantation (BMT) evaluation.

After 15 days from her admission, she developed fever and a broad-spectrum antibiotic therapy with the addition of caspofungin was administered. Blood cultures were positive for *Sphingomonas paucimobilis* and then for *Enterococcus faecium.* In August 2020, she newly became febrile. Blood examination showed again neutropenia and thrombocytopenia (1.04 × 10^3^ and 31 × 10^3^/μL, respectively) and blood cultures were positive for *Saprochaete capitata*. Ceftazidime/avibactam (2.5 g three times a day) and liposomal amphotericin B (3 mg/kg/daily) were started. Voriconazole was associated (6 mg/Kg every 12 h on day 1, then 4 mg/Kg twice a day). Blood cultures became negative after two days, but due to deterioration of the patient’s condition, she was moved to the Intensive Care unit. Blood cultures remained negative but the fungus was again isolated from a sacral lesion despite the antifungal therapy with voriconazole (200 mg twice a day) and amphotericin B was continued. The patient developed septic shock and died in September 2020 from multiple organ dysfunction syndrome in complete remission. Blood cultures were negative both for *Saprochaete capitata* and multiresistant bacteria, even if the fungus persisted in the skin.

Case 2. The second case was documented in a 71-year-old woman with a diagnosis of acute myeloid leukemia with translocation t (2.11) and a mutation in the RUNX1 gene in October 2019. After two months from the allogeneic bone marrow transplantation, the patient presented at the Hematology unit on 21 September 2020 with diarrhea and persistent pain to the right hemi thorax. An antibiotic therapy with piperacillin/tazobactam (4.5 g three times a day) was started with an improvement of the symptoms. The results of a computed tomography showed the presence of a lobar pneumonia with a suspected fungal etiology. After 9 days from admission, the patient had a febrile episode, so piperacillin/tazobactam was empirically replaced with meropenem (1 g three times a day), and an antifungal treatment with caspofungin (70 mg on day 1, followed by 50 mg/daily) was initiated. Blood cultures were positive for *Saprochaete capitata*. At that time, the patient showed a severe neutropenia and thrombocytopenia (1.13 × 10^3^ and 21 × 10^3^/μL, respectively). After fungus identification, caspofungin was switched to liposome amphotericin B (3 mg/kg/daily). For an increase in the creatinine values, liposome amphotericin B was replaced with voriconazole (200 mg twice a day) and anidulafungin (200 mg on day 1, followed by 100 mg/daily). Voriconazole blood levels were checked weekly to target 4000 ng/mL. Blood cultures persisted positive for the fungus until 2 November 2020. 

After 9 November 2020 the patient show an improvement of her clinical condition. The patient left the unit on 20 November 2020 with voriconazole 400 mg three times a day as secondary prophylaxis. The patient recovered completely, but unfortunately died in September 2021 as a consequence of acute myeloid leukemia relapse. No clinical or microbiological reactivation of *Saprochaete capitata* has been highlighted. 

## 2. Microbiological Results

Blood samples were cultivated in the BACTEC system using media for aerobic bacteria (Becton Dickinson Diagnostic Instrument System, Milan, Italy). Subcultures were performed on 5% sheep blood agar and Sabouraud dextrose agar media after positive signaling and microscopic examination of the BACTEC bottles. Colonies grew for 24 h on Sabouraud agar plates at 37 °C. After 48 h of incubation at 37 °C, the macroscopic aspect of white to cream cottony colonies can be observed (Figure 1A).

Light microscopy revealed numerous arthroconidia both in fresh (Figure 1B) and in Gram-stained (Figure 1C) 48 h-old cultures. The isolates were identified by MALDI-TOF mass spectrometry (BioMérieux, Florence, Italy). 

The minimum inhibitory concentration (MIC) of the different antifungal drugs was determined using the Sensititre YeastOne^®^ Y10 panels (Thermo Fisher Inc., Milan, Italy) according to the manufacturer’s instructions. The results are shown in Table 1. 

The MICs for echinocandins (anidulafungin and caspofungin) were the same for the two isolates (2 and 8 μg/mL, respectively), while they were different for micafungin (2 and 8 μg/mL for the two isolates, respectively). Amphotericin B has a MIC of 1 μg/mL in both isolates. High MICs were found for 5-Fluorocytosine (16 μg/mL) and fluconazole (8 μg/mL) and much lower for itraconazole (0.25 μg/mL and 0.5 μg/mL, respectively), posaconazole (0.5 μg/mL and 1 μg/mL, respectively) and voriconazole (0.12 μg/mL and 0.5 μg/mL, respectively).

## 3. Discussion

Rare fungi as arthroconidial yeasts are diffuse globally, even if infections due to *Trichosporon* are prevalently reported in the United States, while those determined by *Saprochaete capitata* and *Saprochaete clavata* were mostly found in the Mediterranean area [8,19,29,33,34,35,36,37,38,39,40]. This underlines that climatic factors might have a role in determining the epidemiology of these pathogens. Clinically, the infections associated with these two species were not different. Regarding diagnostic tests, positive blood cultivation and successive identification by MALDI-TOF or molecular methods remains indicative for this mycosis. It is difficult to establish an early diagnosis, and is very important to start an early antifungal treatment before the development of a more advanced stage of the disease that is more difficult to control. The European Organization for Research and Treatment of Cancer/Mycoses Study Group has proposed β-glucan, a cell wall polysaccharide found in several fungi as an early marker for diagnosing invasive fungal disease [37]. In our cases, we did not find any positivity for galactomannan, another fungal cell wall component against which *Saprochaete capitata* shows cross-reactivity.

Different studies have reported infections with *Saprochaete capitata.* Most of them were associated with patients who had acute myeloid leukemia [8,20,36,41,42]; only a few cases were related to chronic obstructive pulmonary disease, diabetes, solid tumors, solid organ transplantation, or endocarditis [8,36,43].

A clinical picture of *Saprochaete capitata* infections is fungemia, with visceral diffusion to the liver, spleen, kidney, lung and central nervous system [26,28,36,44]. The most probable risk factor of acquiring an opportunistic infection with *Saprochaete capitata* is the neutropenia due to cytotoxic chemotherapy [20,36]. Additionally, our patients presented severe neutropenia at the time of fungemia diagnosis. The use of vascular catheter and antibiotics are the other predisposing conditions that favor *Saprochaete capitata* infections.

In neutropenic patients, *Saprochaete capitata* gives rise to infection development that resembles production by candidiasis and other invasive fungal infections starting with colonization and then proceeding with dissemination and organ involvement. One patient (case 2) of our study had probable pulmonary localization on CT scan even if lung biopsy was not performed. Skin involvement of disseminated *Saprochaete capitata* infection is also frequent and one of our patients (case 1) had a sacral lesion. No guideline establishes the optimal antifungal treatment for *Saprochaete capitata* infections. Arendrup et al. [14] advised the use of amphotericin B (with or without flucytosine) for its prophylaxis; further, another study [36] demonstrated that echinocandins are not the optimal therapeutic choice for the treatment of infections determined by these arthroconidial yeasts.

Clinical breakpoints for *Saprochaete capitata* are not available; however, antifungal susceptibility assays are recommended on all isolates [45]. *Saprochaete capitata* is considered intrinsically resistant to echinocandins; in particular, it was recently found that an F-to-L amino acid substitution in a highly conserved position of the catalytic subunit of β-1,3-d-glucan synthase, Fks1p, might be related to a reduced susceptibility to echinocandins [46].

In our observation, triazoles (except for fluconazole) showed potent in vitro activity against *Saprochaete capitata* with low MIC values. This is in agreement with data reported in the literature, which found that itraconazole, voriconazole and posaconazole (MIC ranges, 0.12 to 0.5, 0.25 to 0.5, and 0.03 to 0.5 μg/mL, respectively) have low MICs against *Saprochaete capitata*, but not fluconazole [17,36,46].

In addition, amphotericin B is reported as the most active drug against *Saprochaete capitata*; however, in this study we found a MIC value higher (1 mg/mL) than that found in another study [32]. The use of the only amphotericin B might explain the persistence of the fungus in sites other than blood (case 1); while in the other patient (case 2) a good resolution of the infection was obtained with the addition of voriconazole.

According to the ESCMID/ECMM guidelines, the suggested regimens are based on amphotericin B with or without flucytosine, but in our cases flucytosine had a MIC value of 16 mg/L, which is much higher than those reported in the literature, and our effective therapy was based on a combination of amphotericin B and voriconazole.

## 4. Conclusions

In conclusion, our cases further underline, together with others, that *Saprochaete capitata* infections represent an emerging problem in the setting of hematological diseases. Then we confirm the feasibility of voriconazole therapy in its treatment co-administered with amphotericin B.

Successful management of this fungal infection is based both on the administration of effective antifungal therapy and on the control of the underlying conditions. In fact, hematological neoplasia and the severity of events leading to the Intensive Care Unit were significantly associated with high rate of mortality. Therefore, it is difficult to establish whether the occurrence of a disseminated fungal infection worsens the prognosis or is a severity marker of the underlying disease. This was the case observed for our first patient, where the cause of her death was probably not directly linked to the fungus infection, which was not more present in blood but in skin lesion.

Clinicians should be aware of this possible infection mostly in neutropenic patients, especially in those with acute myeloid leukemia, with a long history of neutropenia and antifungal prophylaxis or treatment. Prolonged exposure to antimicrobial drugs is associated with breakthrough fungal infections; the management of these fungal infections is complex and requires a collaboration between clinicians and microbiologists. Rapid identification and beginning the appropriate treatment early can, in fact, reduce the high mortality rate associated with these infections, preventing the advanced level of infection and immunosuppression of the patients that may lead to the lack *in vivo* of antifungal drug’s effectiveness.

## Figures and Tables

**Figure 1 vaccines-09-01289-f001:**
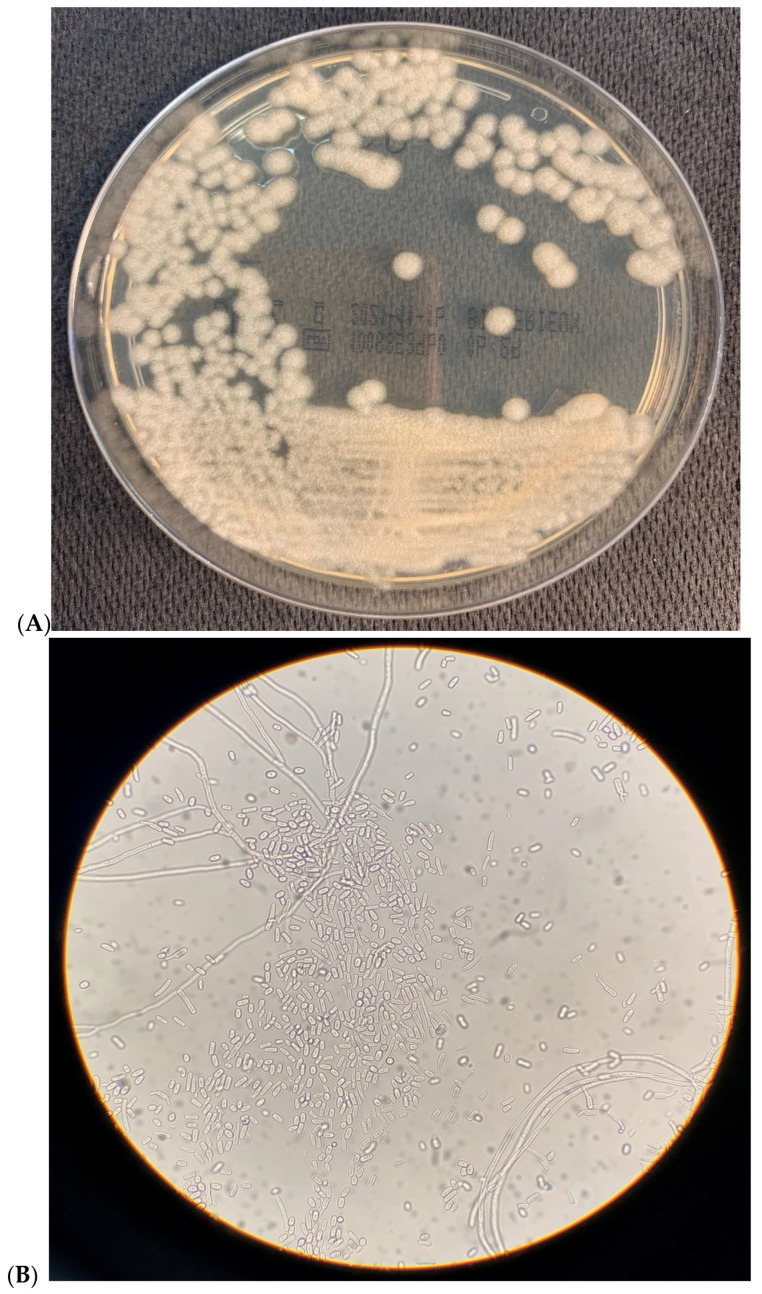
(**A**) Macroscopic aspect of *Saprochaete capitata* in Sabouraud agar plate after 48 h of incubation at 37 °C, showing smooth cream color colonies. (**B**) Light microscopy observation of *Saprochaete capitata* in fresh samples. Numerous arthroconidia are observed. (400× amplification). (**C**). Light microscopy observation of *Saprochaete capitata* in Gram-stained samples. Numerous arthroconidia are observed. (400× amplification).

**Table 1 vaccines-09-01289-t001:** Antifungal susceptibility of *Saprochaete capitata* isolates.

Drugs (μg/mL)	Patient 1	Patient 2
5-Fluorocytosine	16	16
Amphotericin B	1	1
Anidulafungin	2	2
Caspofungin	8	8
Fluconazole	8	8
Itraconazole	0.25	0.5
Micafungin	2	8
Posaconazole	0.5	1
Voriconazole	0.12	0.5

## Data Availability

Data sharing not applicable. No new data were created or analyzed in this study. Data sharing is not applicable to this article.

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
