# Peer review of "Management of Invasive Infections due to a Rare Arthroconidial Yeast, Saprochaete capitata, in Two Patients with Acute Hematological Malignancies"

_vaccines, 2021, doi:10.3390/vaccines9111289_

Round 1

Reviewer 1 Report

In this manuscript, Gurrieri et al. reported two cases of patients with acute myeloid leukemia (AML) that had Saprochaeate capitata infections. Despite similar anti-fungal treatment, one patient had fatal outcome. This report may raise the attention for monitoring of Saprochaeate capitata and other fungal infections in AML patients. But this manuscript has following issues needs to be addressed first before considering for publications.

  1. This manuscript is a case report, I think the authors should address this in the title.
  2. There are some spelling errors in this manuscript, such as CD11b in line 78, is positive or negative? Also untile in line 81?
  3. According the description of case 1, it’s hard to conclude that patient 1 did die from Saprochaeate capitata infection. Especially patient 1 seems react better to anti-fungal drugs.
  4. I think the authors need to show more details or at least discuss more about the cause of death of both patients.
  5. In conclusion part, I don't feel comfortable to refer the work in this manuscript a study, is more to be described as observation and reporting.

Author Response

Reviewer 1

In this manuscript, Gurrieri et al. reported two cases of patients with acute myeloid leukemia (AML) that had Saprochaeate capitata infections. Despite similar anti-fungal treatment, one patient had fatal outcome. This report may raise the attention for monitoring of Saprochaeate capitata and other fungal infections in AML patients. But this manuscript has following issues needs to be addressed first before considering for publications.

  1. This manuscript is a case report; I think the authors should address this in the title.

Response 1: We agree with the reviewer’s suggestion and we have inserted the number of cases described in the title.

  1. There are some spelling errors in this manuscript, such as CD11b in line 78, is positive or negative? Also “untile” in line 81?

Response 2: All the spelling errors have been corrected.

  1. According the description of case 1, it’s hard to conclude that patient 1 did die from Saprochaeate capitata infection. Especially patient 1 seems react better to anti-fungal drugs.

Response 3: We agree with the reviewer’s comment about the role that the fungus played as a cause of the patient’s death and we have inserted some sentences in the Conclusions section.

  1. I think the authors need to show more details or at least discuss more about the cause of death of both patients.

Response 4: We have added more information about the cause of death of both patients.

In conclusion part, I don't feel comfortable to refer the work in this manuscript a study, is more to be described as observation and reporting.

Response: We agree with the comment of the referee and we have deleted the term “study”  reported in the paper.

Reviewer 2 Report

The authors describe two cases of sepsis due to Saprochaetae capitatus infection in two patients with acute leukemia. Although the cases may be interesting, they do not add anything special to the current knowledge in terms of prevention, diagnosis and/or management of the disease.

Author Response

Reviewer 2

Comments and Suggestions for Authors

The authors describe two cases of sepsis due to Saprochaetae capitatus infection in two patients with acute leukemia. Although the cases may be interesting, they do not add anything special to the current knowledge in terms of prevention, diagnosis and/or management of the disease

Response: We agree with the comment of the referee 2. We know that our cases are similar to those already reported in literature. However, we believe that to communicate all the cases that occur is very important by an epidemiological point of view because it may give an information about the increased incidence of these rare fungal infections in hematological populations in particular in our geographical area where Saprochaete spp. seem more diffuse for climatic factors. This might alert clinicians to have a clinical suspicion when they manage neutropenic patients, especially those with AML, which remain febrile even if they are treated with echinocadins in order to timely administrate a targeted antifungal therapy.

Reviewer 3 Report

The authors underline the pathogenic role of Saprochaete capitata in neutropenic patients and the absence of guideline to establish the optimal antifungal treatment for infections related to this microrganism.              Some changes are needed, there are spelling and punctuation errors:   

review the spelling and the punctuation;                                                         revise the name of microrganisms, modify Saprochaetae capitatus in Saprochaete capitata throughout the paper and all the other incorrect names of microrganisms also throughout the references, for example C. aruris (It is C. auris).

The microbiological results need some additional information:                       include the conditions of time and temperature for microbial growth;          clarify  the condition of  Fig.1A  described,  on line 133, as "White to cream cottony colonies grew in 24h on agar plates," and differently described on line 136 as "Macroscopic aspect of Saprochaete capitata in Sabouraud agar plate after 48h of incubation at 37°C, showing smooth cream color colonies";                                                                                               

confirm if  the two microscopic observations are made from the same sample and at the same time;                                                                            amplify the epidemiological data including  in the introduction 

Author Response

Reviewer 3

The authors underline the pathogenic role of Saprochaete capitata in neutropenic patients and the absence of guideline to establish the optimal antifungal treatment for infections related to this microrganism.              Some changes are needed, there are spelling and punctuation errors:   

  1. review the spelling and the punctuation;

Response 1: All the mistakes have been corrected 

  1. revise the name of microrganisms, modify Saprochaetae capitatusin Saprochaete capitata throughout the paper and all the other incorrect names of microrganisms also throughout the references, for example  aruris (It is C. auris).

Response 2: All the names of the microorganisms have been revised.

The microbiological results need some additional information:                     

  1. include the conditions of time and temperature for microbial growth;       

Response 1: We have added the conditions of time and temperature for microbial growth

  1. clarify  the condition of  Fig.1A  described,  on line 133, as "White to cream cottony colonies grew in 24h on agar plates," and differently described on line 136 as "Macroscopic aspect of Saprochaete capitata in Sabouraud agar plate after 48h of incubation at 37°C, showing smooth cream color colonies";  

Response 2: Colonies grow and are visible on the Sabouraud agar plates after 24 hours. To observe better their macroscopic and microscopic aspect we performed photographs after 48hours

  1. confirm if  the two microscopic observations are made from the same sample and at the same time;

Response 3: We confirm that the observations were made from the same samples and at the same time

  1. amplify the epidemiological data including  in the introduction

Response 4: We have added more epidemiological data in the Introduction section

Round 2

Reviewer 2 Report

The authors have addressed the comments of the other Reviewers; unfortunately, the main concerns described in my previous report remain.

Author Response

The authors have addressed the comments of the other Reviewers; unfortunately, the main concerns described in my previous report remain.

Response: The referee 2 gives just a comment about this paper without any suggestion to improve it. We agree with his opinion about the addition of these to other similar case reports present in the literature. However, besides what we reported in our first reply, we think that our case reports might give information also about different MIC values for some drugs that in literature are reported as effective.

The in vitro study of susceptibility of S. capitata is quite weak (Duan et al., 2019). Published literature showed that S. capitata is susceptible to flucytosine (MIC values 0.25-0.5 mg/mL), itraconazole (MIC values 0.12-0.50 mg/mL), voriconazole (MIC values 0.25-0.5 mg/mL) and posaconazole (MIC values 0.03-0.25 mg/mL).

Then, in a study of Tortorano et al. (Medical Mycology, 2019), the authors reported that posaconazole MIC90 of 1–2 mg/l might explain the occurrence of breakthrough infections under this triazole reported in the FungiScope registry with a MIC value for this drug different from that reported by the previous study . MIC90 of amphotericin B, itraconazole, and voriconazole maintained a MIC < 1mg/l. In our cases, posaconazole had a MIC of 0.5 and 1mg/l, while amphotericin B MIC was higher (1mg/l). According to the ESCMID/ECMM guidelines, the suggested regimens is based on amphotericin B with or without flucytosine, but in our cases flucytosine had a MIC value of 16mg/l, that is much higher than those reported in the literature and our effective therapy was based on a combination of amphotericin B and voriconazole. This sentence has been added in the Discussion section

So we think that the monitoring  of antifungal susceptibility and reporting it is very important to alert for eventual change of MICs that might condition the effective therapy.